# Development of a Multidisciplinary Aerodigestive Program: An Institutional Experience

**DOI:** 10.3390/children8070535

**Published:** 2021-06-23

**Authors:** Seung Kim, Mireu Park, Eunyoung Kim, Ga Eun Kim, Jae Hwa Jung, Soo Yeon Kim, Min Jung Kim, Da Hee Kim, Sowon Park, Hong Koh, In Geol Ho, Seung Ki Kim, Sangwon Hwang, Kyeong Hun Shin, Hosun Lee, Bobae Lee, Hyeyeon Lee, Minhwa Park, Myung Hyun Sohn, Dong-wook Rha, Kyung Won Kim

**Affiliations:** 1Division of Gastroenterology, Hepatology and Nutrition, Department of Pediatrics, Yonsei University College of Medicine, Severance Children’s Hospital, Seoul 03722, Korea; pedks@yuhs.ac (S.K.); SOWON81@yuhs.ac (S.P.); khong@yuhs.ac (H.K.); LBB1984@yuhs.ac (B.L.); 2Department of Pediatrics, Severance Hospital, Institute of Allergy, Institute for Immunology and Immunological Diseases, Severance Biomedical Science Institute, Brain Korea 21 PLUS Project for Medical Science, Yonsei University College of Medicine, Seoul 03722, Korea; QKRALFM27@yuhs.ac (M.P.); GAEUNKIM@yuhs.ac (G.E.K.); JUNG627B@yuhs.ac (J.H.J.); SOPHI1@yuhs.ac (S.Y.K.); MJ1217@yuhs.ac (M.J.K.); MHSOHN@yuhs.ac (M.H.S.); 3Department of Nursing, Severance Children Hospital, Yonsei University Health System, Seoul 03722, Korea; DLDHTKATK@yuhs.ac; 4Department of Otorhinolaryngology, Yonsei University College of Medicine, Seoul 03722, Korea; DHK@yuhs.ac; 5Department of Pediatric Surgery, Severance Children’s Hospital, Yonsei University College of Medicine, Seoul 03722, Korea; HNJKLOP@yuhs.ac; 6Department of Rehabilitation Medicine, Yongin Severance Hospital, Research Institute of Rehabilitation Medicine, Yonsei University College of Medicine, Seoul 03722, Korea; GSG1230@yuhs.ac; 7Department and Research Institute of Rehabilitation Medicine, Yonsei University College of Medicine, Seoul 03722, Korea; ALWAYSWON87@yuhs.ac (S.H.); MEDICUS@yuhs.ac (D.-w.R.); 8Department of Nutrition Care, Severance Hospital, Yonsei University Health System, Seoul 03722, Korea; KHSHIN84@yuhs.ac (K.H.S.); HSLEE0730@yuhs.ac (H.L.); 9Department of Pediatric Occupational Therapy, Severance Rehabilitation Hospital, Yonsei University Health System, Seoul 03722, Korea; FEELING222@yuhs.ac (H.L.); ALSGHKWOW@yuhs.ac (M.P.)

**Keywords:** aerodigestive team, children, feeding, multidisciplinary care, pediatric, respiratory difficulty, swallowing

## Abstract

We share our experience on the implementation of a multidisciplinary aerodigestive program comprising an aerodigestive team (ADT) so as to evaluate its feasibility. We performed a retrospective chart review of the patients discussed at the monthly ADT meetings and analyzed the data. A total of 98 children were referred to the ADT during the study period. The number of cases increased steadily from 3.5 cases per month in 2019 to 8.5 cases per month in 2020. The median age of patients was 34.5 months, and 55% were male. Among the chronic comorbidities, neurologic disease was the most common (85%), followed by respiratory (36%) and cardiac (13%) disorders. The common reasons for consultation were suspected aspiration (56%), respiratory difficulty (44%), drooling/stertor (30%), regurgitation/vomiting (18%), and feeding/swallowing difficulty (17%). Following discussions, 58 patients received active interventions, including fundoplication, gastrostomy, laryngomicrosurgery, tracheostomy, and primary dilatation of the airway. According to the questionnaire of the caregiver, the majority agreed that the main symptoms and quality of life of patients had improved (88%), reducing the burden on caregivers (77%). Aerodigestive programs may provide comprehensive and multidisciplinary management for children with complex airway and digestive tract disorders.

## 1. Introduction

As medical science advances and new subspecialties arise, there is a renewed demand for multidisciplinary team approaches (MTAs) to manage patients with complicated multisystem medical problems. There is compelling evidence supporting the effectiveness of MTAs in improving clinical outcomes, decreasing adverse events, increasing the satisfaction of patients and healthcare providers, and reducing the burden of caregivers [1,2].

A pediatric aerodigestive program is a multidisciplinary program aimed to diagnose and to treat pediatric patients with complex multi-systematic problems affecting their airways, breathing, feeding, swallowing, and growth [3]. The aerodigestive tract comprises the organs of the respiratory tract and the upper part of the digestive tract including the pharynx, larynx, trachea, and esophagus. As these organs are adjoined and are involved in swallowing and breathing, structural integrity and complex coordination are essential for their proper functioning. Since children are constantly growing and developing, the aerodigestive tract problems in children are challenging to handle [4]. Even an experienced specialist is likely to look at the problem from their own field and to overlook problems that could be associated with other disciplines when they see a patient. A lack of seamless coordination among several medical specialists can increase the efforts, time, and cost of proper evaluation or intervention.

A pediatric aerodigestive team (ADT) is necessary for the high level of coordination and integrated team effort needed to provide efficient and higher quality care [3].

Several researchers have reported positive clinical outcomes in patients treated by ADTs [5,6]. Furthermore, it has been reported that ADTs can reduce unnecessary tests, medical costs, risks, and time for diagnoses [7,8]. The first aerodigestive multidisciplinary program was initiated at Cincinnati Children’s Hospital in the United States in 1999 [9]. Since then, pediatric aerodigestive programs have become more prevalent, with approximately 50 aerodigestive centers established across the United States of America [3,10]. However, aerodigestive programs are still scarce in Asia, even in advanced tertiary care setup.

Our institute is one of the biggest tertiary referral centers in South Korea. We initiated a pediatric aerodigestive program two years ago, and we are still in the process of establishing a few components. In this paper, we share our experience and the progress of implementing an ADT in our institute.

## 2. Materials and Methods

### 2.1. Initiation and Establishment of ADT

The first meeting for developing an ADT was held in July 2018. Pediatric pulmonologists, gastroenterologists, nurses, a rehabilitation team (i.e., physicians and occupation therapists) for swallowing evaluation and rehabilitation, and pediatric surgeons participated as the initial team members. After multiple meetings, discussions, and literature reviews, including referring to websites of well-established ADTs in the United States, the ADT development was initiated in Severance Children’s Hospital in November 2018. 

Monthly meetings were held for case discussions. During ADT consultations, thorough patient information was shared among ADT members. If required, specialists from different disciplines examined or evaluated the patient before the meeting after reviewing their medical records. In an outpatient case, an ADT nurse arranged the appointment with several departments to minimize the number of hospital visits and financial burden. 

Nutritionists and pediatric otolaryngologists joined as team members shortly after ADT commencement. As an ADT was not a familiar concept for patients and medical staff, we provided medical staff with ADT information via email. We also placed posters to inform patients, parents, and medical staff about the ADT. In the email and educational posters, we described the purpose of an ADT, the members of our ADT, and their roles. We also specified indications for consultation, namely patients with problems including chronic respiratory difficulty, swallowing difficulty, and regurgitation or aspiration. 

### 2.2. Study Population and Data Collection

A retrospective chart review was conducted for patients whose cases were discussed at the monthly ADT meetings of Severance Children’s Hospital from November 2018 to June 2020. All patients who were referred to the ADT during the study periods were enrolled in the study. Patient information, including underlying medical conditions, feeding and respiratory status, chief complaint at consultation, and anthropometric data, were reviewed and collected. Data regarding evaluations and interventions after ADT were also collected. For the patients who received an intervention after the ADT meetings, a caregiver completed a simple questionnaire to collect feedback (Appendix A) after three months of ADT intervention. Two years after the initiation of the ADT, we shared a questionnaire with the medical staff who participated in the ADT to evaluate the impact of ADT activity on their medical services and attitude toward the medical issues of patients (Appendix A).

### 2.3. Statistical Analyses and Ethics

After performing the Kolmogorov–Smirnov test, patient age and anthropometric data were expressed as the medians and interquartile ranges (as those were not normally distributed). A trend test was performed using the chi-square test. Statistical analyses were performed using SPSS software version 18.0 (SPSS Inc., Chicago, IL, USA) and R (R Foundation for Statistical Computing, Vienna, Austria; version 4.0.1). 

This study was reviewed and approved by the Institutional Review Board of Yonsei University Health System, Severance Hospital (approval date 22 June 2020; approval No. 4-2020-0493).

## 3. Results

Figure 1 shows a timeline of establishing the ADT and monthly cases needing ADT consultation. Although there were variations between months, consultation cases increased steadily as the ADT settled in. The mean monthly cases in 2018, 2019, and 2020 were 0.7, 3.8, and 11.8, respectively.

A total of 98 cases were referred to the ADT during the study period. The median age of patients was 34.5 months, and 55 (56.12%) patients were male. All included patients had more than one underlying chronic comorbidity. Among them, neurologic diseases were the most common (84.69%), followed by respiratory (35.71%) and cardiac (13.26%) disorders. The most common reasons for consultation were suspected aspiration (56.12%), respiratory difficulty (43.88%), drooling or stertor (29.59%), regurgitation or vomiting (18.37%), and feeding or swallowing difficulty (17.35%). Most patients had a failure to thrive, which could represent insufficient nutritional status. Among the enrolled patients, 13 (13.26%) were on respiratory support, and 58 (59.18%) were receiving tube feeding. Fifty-nine (60.20%) of the consulted patients were inpatients, and six of them were in the pediatric intensive care unit (PICU) (Table 1).

We counted the total cases of aerodigestive tract-associated evaluations and interventions performed at our institute during the study period. After analyzing the numbers of ADT-related evaluations, all but chest computed tomography (CT) showed increasing trends since the initiation of the ADT. Regarding ADT-related interventions, the cases of tracheostomy and gastrostomy did not show significant differences before and after the ADT. However, fundoplication, laryngomicrosurgery, and airway bougination showed an increasing trend. Furthermore, intriguingly, among all performed tests, percentages of tests conducted based on ADT recommendation had increased when we compared the first and second year after the ADT was established (Table 2). Among the 98 enrolled patients, after in-depth reviews and discussions, 58 patients received invasive interventions, including fundoplication surgery (*n* = 23, 39.7%), gastrostomy(*n* = 18, 31.0%), laryngomicrosurgery (*n* = 10, 17.2%), tracheostomy (*n* = 4, 6.9%), and primary dilatation of the airway (*n* = 3, 5.2%). According to the questionnaire given to caregivers (*n* = 45), a majority answered that the main symptoms of consultation and quality of life had remarkably improved (88%), resulting in reduction of caregiver burden (77%). 

The questionnaire that was administered to medical staff who were participating in ADT as team members was completed two years after initiating ADT formation. Surprisingly, 56% of medical staff were not even aware of the role and/or presence of an ADT before their participation. Furthermore, 67% of staff responded that their diagnostic or therapeutic approach had changed significantly after ADT activities. All responders agreed that their understanding of patient conditions had improved along with the quality of treatment. Additionally, through ADT meetings, all medical specialists agreed that their understanding of other specialty fields of ADT had improved not only in terms of medical knowledge but also focusing on points and hurdles of patients care. However, all medical staff answered that they were devoting more time to each patient, which resulted in an increased workload. 

## 4. Discussion

As the establishment of ADTs is becoming more common, an academic society that exclusively deals with pediatric aerodigestive diseases has recently emerged, and a consensus statement has been announced regarding the structure and functions of pediatric aerodigestive programs in 2018 [3]. Recently, Lindsey et al. investigated the size and prevalence of pediatric aerodigestive programs by conducting an email survey of international pediatric gastroenterologists [10]. According to the report, 34 centers responded, and most (88.2%) were located in the United States. Only four (11.8%) centers were located outside of the United States (one in Canada, two in Europe, and one in Central America). Currently, there are no reports regarding pediatric ADT from Asia. Furthermore, the concept of an ADT is still unfamiliar in South Korea, even among medical personnel. In our survey, more than half of the ADT members were not aware of the term or the role of an ADT before joining the team. This is the first study to introduce the adoption process of pediatric ADT, and we hope our experience can serve as a reference for the establishment of ADTs. In our ADT, monthly consultation cases increased gradually from 0.7 to 11.8 over the first two years. This result is concordant with a previous study that reported a positive correlation between the duration of aerodigestive programs and number of patients. These results suggest that there is a hidden demand for ADT services which often goes unnoticed or overlooked.

Although there is a consensus statement on pediatric aerodigestive programs, a standardized structure or guideline has not yet been developed, and there are significant variations in the operation of aerodigestive programs among hospitals [3,10]. Medical services in South Korea are under a national insurance coverage system, and in this system hospitals cannot impose multidisciplinary ADT fees on patients. Thus, it was impossible to set up a clinic dedicated to ADT patients. As a result of this non-conducive environment, our team shared patient data before meetings and could only perform monthly meetings during our lunch breaks for the most effective and practical team meetings. Medical systems may differ by country, and ADT operations should be customized according to the circumstances and regulations of relevant institutes. However, to be the most effective team, patient and family-focused face-to-face multidisciplinary clinics should be pursued in the future [3].

A total of 58 patients underwent invasive surgical treatment through our aerodigestive program. Some patients received additional medical interventions such as swallowing rehabilitation, pharmacologic treatment, and dietary modification through advice from various specialists, which was possible through patient-centered in-depth discussion among team members. In this study, among the 98 enrolled patients, 59 (60.2%) were inpatients, including six patients in the PICU, and 84.69% of patients had underlying neurologic disorders. We hope that patient scope can be expanded to outpatient clinics and include other indications such as craniofacial malformations or chronic cough [4]. Furthermore, premature birth is among the well-known risk factors of an aerodigestive problem [11]. Also, in our study patients, 29 (29.6%) patients had a history of premature birth. For these reasons, neonatologists have also been included in ADTs since April 2020. Furthermore, pediatric cardiologists have also joined ADTs since July 2020 for precise decision making for relevant patients.

Upon comparing the pre and post-ADT eras, related evaluations and interventions showed an increasing trend in general. Notably, bronchoscopy, laryngomicrosurgery, and bougination have increased drastically since the second year of the ADT. This might be due to the higher number of active interventions as a part of the aerodigestive program. The management strategy or indications of evaluations might have changed through in depth discussion with other specialists. These might also have changed due to the increase in the number of patients who were overlooked for aerodigestive evaluation and/or management before the ADT. Although it is known that ADTs might help reduce unnecessary tests, we believe that our increasing trend is reflecting an increase in “necessary tests”. However, as our process of setting up an ADT is still evolving, we have to follow the changing trends.

Furthermore, an increasing percentage of ADT consultations among those who received interventions suggests that the role of ADTs is becoming more established. Triple scope procedure (flexible bronchoscopy, rigid bronchoscopy, and gastroesophagoduodenoscopy) is regarded as one of the merits of an aerodigestive program that can help reduce unnecessary anesthesia and medical costs while increasing diagnostic yield and patient satisfaction [7]. However, our ADT was not yet prepared to perform triple scope due to practical hurdles that must be overcome. Although it is well studied that MTA is effective in treating complicated medical issues, this approach is focused on intensive care, which requires considerable human resources. To be sustainable, medical personnel and systematic and institutional support should be guaranteed. Furthermore, medical specialists who are exclusively involved with ADTs, such as a coordinator, are necessary for the ideal organization of ADTs.

The questionnaire administered to the ADT members revealed that the concept of an ADT is still unfamiliar in Asia, even among pediatric specialists working at tertiary referral centers. Although ADTs are becoming more common globally, most aerodigestive programs are located in North America, and there is a need to introduce and expand these programs to other parts of the world. All participating medical personnel agreed to participate in improving the quality of care, but this resulted in increased workload. 

There are some limitations to this study that must be considered while interpreting our results. First of all, we could not evaluate the objective clinical outcome of the ADT due to the short follow-up period and evolving condition of the ADT. Detailed clinical follow up, including nutritional status, should be performed to maximize the effect of the ADT. Although we have received positive feedback on the ADT from the caregivers of consulted patients, we used simple non-validated questionnaires in this study, and as there are no data for the comparison with the pre-ADT era, it is difficult to conclude that patient satisfaction has been raised by the ADT itself. Prospective long-term studies with validated questionnaires are needed.

## 5. Conclusions

We have successfully initiated our ADT, which is still evolving and developing. Our ADT initiated a change in clinical practice patterns toward more active evaluations and management of the patients who require aerodigestive care. We share our experience in this paper, and we expect that this can help other care centers establish ADTs similar set in their institutions, which is tailored according to the needs and situations of each hospital. 

## Figures and Tables

**Figure 1 children-08-00535-f001:**
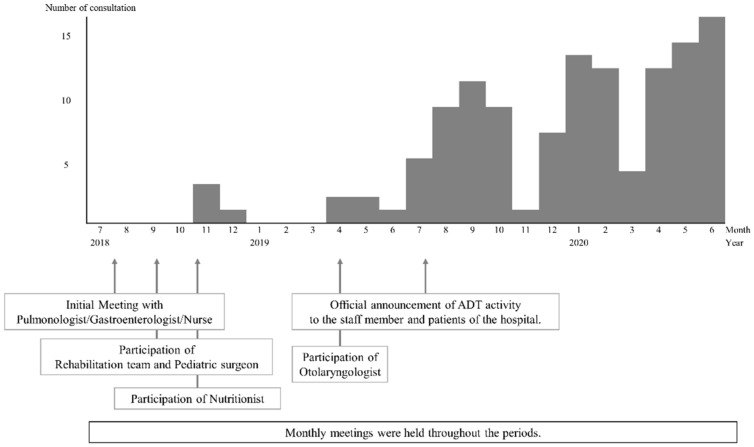
A timeline of ADT establishment and the monthly number of consultations.

**Table 1 children-08-00535-t001:** Subject characteristics at the initial consultation (N = 98).

Variables	Values
Age (month)	34.50 (13.00, 90.75)
Sex, male	55 (56.12)
Anthropometric data	
Weight Z-score	−2.44 (−3.64, −1.13)
Height Z-score	−2.04 (−3.44, −0.78)
Weight for height/BMI Z-score ^†^	−1.02 (−2.58, 0.16)
Primary medical condition *	
Neurologic disorder	83 (84.69)
Pulmonological disorder	35 (35.71)
Gastroenterological disorder	6 (6.12)
Cardiac disorder	13 (13.26)
Others (malignancy, skeletal disorders, etc.)	7 (7.14)
Reason for consultation *	
Respiratory difficulty	43 (43.88)
Aspiration	55 (56.12)
Feeding or swallowing difficulty	17 (17.35)
Regurgitation, vomiting	18 (18.37)
Drooling, stertor	29 (29.59)
Others	8 (8.16)
Respiratory support	
None	85 (86.73)
O_2_	6 (6.12)
Mechanical ventilation	7 (7.14)
NIV (non-invasive ventilation)	3 (3.06)
IMV(invasive mechanical ventilation)	4 (4.08)
Tracheostomy	6 (6.12)
Feeding route	
Oral	40 (40.82)
Naso(oro)gastric tube	34 (34.69)
Gastrostomy	24 (24.49)
Patient status	
Outpatient	39 (39.80)
Inpatient ^†^	59 (60.20)

Values are presented as median (interquartile range) values or *n* (%). ^†^ Weight for height Z-score was applied for the patients less than two years old, and BMI Z-score was applied for the patients above two years of age. * If a patient had several conditions simultaneously, duplications were allowed.

**Table 2 children-08-00535-t002:** Evaluations and interventions before and after ADT.

	Pre-ADT Era		Post-ADT Era		*p*-Value ^†^
	2016.11–2017.10	2017.11–2018.10	2018.11–2019.10	2019.11–2020.6	
Evaluation, n (% of ADT consulted case)					
Laryngoscopy	774	813	998 (2.2)	417 (10.3)	<0.0001
Chest CT	642	711	632 (5.7)	447 (8.3)	0.1525
Bronchoscopy	43	55	53 (9.4)	82 (41.5)	<0.0001
Esophagography	165	175	257 (11.3)	153 (27.5)	<0.0001
VFSS	115	121	200 (23.0)	127 (37.0)	<0.0001
MII-pH	62	82	109 (26.6)	89 (42.7)	<0.0001
Intervention, n (% of ADT consulted case)					
Tracheostomy	34	27	16 (0.0)	21 (19.1)	0.3207
Gastrostomy	51	44	49 (8.2)	30 (46.7)	0.3723
Fundoplication	38	36	39 (12.8)	39 (46.2)	0.0311
LMS	9	8	15 (13.3)	23 (34.8)	<0.0001
Bougination	1	1	0 (0.0)	11 (27.3)	0.0022

ADT, Aerodigestive team; CT, computed tomography; LMS, Laryngomicrosurgery; MII-pH, multichannel intraluminal impedance-pH monitoring; VFSS, videofluoroscopic swallow study. ^†^ Chi-square test was used for the statistical analysis. Ratio of evaluations and interventions to hospitalized patients number were compared between before and after ADT.

## Data Availability

The data presented in this study are available on request from the corresponding author. The data are not publicly available due to ethical reasons.

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
