# Peer review of "Development of a Multidisciplinary Aerodigestive Program: An Institutional Experience"

_children, 2021, doi:10.3390/children8070535_

Round 1
Reviewer 1 Report
Authors have described their center's experience of establishing ADT.
This is a successful and essential model to be implemented in all tertiary care centers. Overall this is a well written manuscript and I just have few questions for the authors.
In table 2. Of all the evaluations, how many of them have actionable results/positive results. Did that change before and after implementation.
In other words, are we just doing more evaluations when not needed or is it actually helping in medical/surgical management.
Are there objective data like improvement in Z score of patients in ADT or before and after evaluation. Less respiratoy distress or need for hospitalization etc.
Author Response
Response to Reviewer 1 Comments
Point 1: In table 2. Of all the evaluations, how many of them have actionable results/positive results. Did that change before and after implementation.
In other words, are we just doing more evaluations when not needed or is it actually helping in medical/surgical management.
Response 1:
Thank you for your valuable comment. We totally agree with your idea. It is ideal to analyse whether ADT changes the clinical decision for the patient or not. However, using the ratio of a positive result to a total number of evaluations has limitations since before ADT, evaluations might have performed when the patient has definite and severe symptoms, leading to a more positive result. That is why we have focused on the trend of number of evaluations and interventions and the % of ADT consulted cases. We believe that, through ADT, patients with aerodigestive problems are receiving more precise and personalized evaluations and required intervention. We had described this issue in the discussion session as below.
Line: 219-228
Upon comparing the pre- and post-ADT eras, related evaluations and interventions showed an increasing trend in general. Notably, bronchoscopy, laryngomicrosurgery, and bougination have increased drastically since the second year of ADT. This might be due to the higher number of active interventions as a part of the aerodigestive program. The management strategy or indications of evaluations might have changed through in depth discussion with other specialists. These might also have changed due to the increase in the number of patients who were overlooked for aerodigestive evaluation and/or management before ADT. Although it is known that ADT might help reduce unnecessary tests, we believe that our increasing trend is reflecting an increase in “necessary tests.” However, as our process of setting up an ADT is still evolving, we have to follow the changing trends.
Point 2: Are there objective data like improvement in Z score of patients in ADT or before and after evaluation. Less respiratoy distress or need for hospitalization etc.
Response 2: Thank you for the comment. Unfortunately, we could not compare the objective outcome of ADT and described it as our limitation as below.
Line: 247-255
There are some limitations to this study that must be considered while interpreting our results. First of all, we could not evaluate the objective clinical outcome of the ADT due to the short follow-up period and evolving condition of the ADT. Detailed clinical follow up, including nutritional status, should be performed to maximize the effect of the ADT.

Reviewer 2 Report
The concept of “integrated practice unit (IPU)” is the central idea of this paper. This is an evolving field and the difficulties and obstacles faced by MDTs during their development and operation in different healthcare systems are valuable for other scientific teams on the way of establishing MDT practice. This is a topic of great interest regarding both the anticipated improvement of patient care and outcomes and the reduction of healthcare and societal costs and numerous stakeholders in the provision of health care for complex aerodigestive problems can benefit by the discussion of practices applied in existing MDTs. The authors present in the introduction the general international trends in this subject.
Some suggestions to the authors:
Key words: add swallowing, feeding, respiratory difficulty
Line 55: A pediatric aerodigestive program
Line 105: “respiration status” rephrase to “respiratory status”
Line 105: “chief complaint about consultation” rephrase to “chief complaint at consultation”
Line 108: “a caregiver administered a simple questionnaire” rephrase to “a caregiver completed a simple questionnaire”
Line 130: “The common reason for consultation was” rephrase to either “The most common reasons for consultation were” or “Common reasons for consultation were”
Line 153: “gastrostomy formation” rephrase to “gastrostomy”
Table 2: a footnote should explain which comparison the p values correspond to (what test and between which values)
Line 174: “constitution” rephrase to “establishment”
Author Response
Response to Reviewer 2 Comments
Point 1: Key words: add swallowing, feeding, respiratory difficulty
Response 1: Thank you for your suggestion. We have added key words as reviewer’s recommendation.
Point 2: Line 55: A pediatric aerodigestive program
Response 2: We have corrected as reviewers correction.
Point 3: Line 105: “respiration status” rephrase to “respiratory status”
Response 3: We have corrected as reviewers correction.
Point 4: Line 105: “chief complaint about consultation” rephrase to “chief complaint at consultation”
Response 4: We have corrected as reviewers correction.
Point 5: Line 108: “a caregiver administered a simple questionnaire” rephrase to “a caregiver completed a simple questionnaire”
Response 5: We have corrected as reviewers suggestion.
Point 6: Line 130: “The common reason for consultation was” rephrase to either “The most common reasons for consultation were” or “Common reasons for consultation were”
Response 6: We have corrected as reviewers correction.
Point 7: Line 153: “gastrostomy formation” rephrase to “gastrostomy”
Response 7: We have corrected as reviewers suggestion.
Point 8: Table 2: a footnote should explain which comparison the p values correspond to (what test and between which values)
Response 8: We apologize for the insufficient description. We have added foot note for the understanding as below
† Chi-square test was used for the statistical analysis. Ratio of evaluations and interventions to hospitalized patients number were compared between before and after ADT.
Point 9: Line 174: “constitution” rephrase to “establishment”
Response 9: We have corrected as reviewers suggestion.

This manuscript is a resubmission of an earlier submission. The following is a list of the peer review reports and author responses from that submission.
Round 1
Reviewer 1 Report
Authors have described their Aerodigestive team experience in this manuscript.
Materials and Methods:
Authors have mentioned ADT, is it a multidisciplinary clinic where patients see different specialities in a single day which improves the efficiency and also decreases the need for frequent hospital visits or just a multidisciplinary discussion of patients as in conference? This needs to be stated clearly.
Line 79: What does rehabilitation service comprises of? Occupation therapy/physical/feeding team? Please elaborate.
Results: Authors have mentioned multimorbidity among the patients. I am curious not to see prematurity being mentioned who will have both swallowing incordination and also neurological sequelae due to variety of reasons such as IVH among others. Did the authors evaluate for prematurity?
Table 1: Did the authors evaluate MUAC or weight for length/height - these will be better measures of nutritional status than Weight and height. Further did the authors see any improvement in BMI or weight for length Z scores? As improvement in nutritional status can be used as one of the markers of outcomes of Aerodigestive team in terms of better feed tolerance.
Table 2: is a descriptive table and is not useful without doing statistical testing to see if things have improved. Kindly consult a statistician to come up with appropriate indicators for pre and post comparison and testing or trend analysis. Just a higher number is less meaningful if not statistically significant.
Why there is different between pre and post? Previously individual subspecialities were not doing their due diligent?
Regarding the questionnaire, please state how many participated in the survey in terms of absolute number. Can you give us a breakdown of caregiver survey who were inpatient or outpatient? Since there is no comparison, it is difficult to say whether ADT improved outcomes in terms of patient satisfaction.
How do you designed the questions in survey? is it validated to measure the quality of life? This is a vague questionnaire.
Conclusion: Please rewrite the conclusion. Conclusion gives you an opportunity to summarize the striking feature of your research work. There are no objective data to suggest that quality of care or life improved.
Reviewer 2 Report
The concept of “integrated practice unit (IPU)” is the central idea of this paper. This is a topic of great interest regarding both the anticipated improvement of patient care and outcomes and the reduction of healthcare and societal costs.
This is an evolving field and the difficulties and obstacles faced by ADTs during their development and operation in different healthcare systems are valuable for other scientific teams on the same way of MDT practice.
Some suggestions to the authors:
The introduction would benefit by a short presentation of the two operational concepts compared in the article: what is the care of breathing and swallowing/feeding disorders like without and with a ADT.
Page 2 line 53 First paragraph …increase the satisfaction of patients and healthcare providers “and reduce the caregivers’ burden”
Page 2 Line 57 and tissues
Page 2 Line 59 “Structural” instead of morphological integrity
Page 2 Line 71 “even in medically developed countries” rephrase to “even in the context of advanced tertiary care provision”
Page 2 Line 74 Please rephrase “In this paper, ….evaluate its feasibility and effectiveness by analyzing patient outcomes.”
If the authors set about evaluating “the feasibility and effectiveness” they are expected to present their results and comment distinctively for both. Either rephrase this or change the presentation of results and discussion accordingly.
Correct ref 9. Is it “Hartnick et al .Assessing the Value of Pediatric
Aerodigestive Care. NEJM Catalyst Innovations in Care Delivery 2020; 04DOI:https://doi.org/10.1056/CAT.19.1132” ??
Supplementary table 1. Rephrase
“How much do your family satisfy about the aerodigestive program?” to
“How satisfied is your family with the aerodigestive program?”
Supplementary table 2.
“When self-assessed, did the patient's quality of care change before and after ADT?”
There should be some description of what was the self-reporting of the recipients of care
There are indeed many “variations in the operation of aerodigestive programs among hospitals” . The operational characteristics of this ADT work should be described. Did the meetings involve only patient cases discussion or also examination (clinical examination or/and flexible nasolaryngoscopy or FEES as office – procedures)?
Page 5 line 185 “A total of 58 patients underwent invasive surgical treatment though through our aerodigestive program”
A comment is made in the introduction that ADTs can reduce unnecessary tests. The authors should discuss about their increase of investigations.
Line 221 In conclusion,
Funding: Please add: